# A microRNA generated via lysosomal processing of ribosomal RNA suppresses proinflammatory responses

Dan Michael[1,2], Ester Feldmesser[1], K Shanmugha Rajan[3], Yoav Lubelsky[4], Anat Bashan[3], Alexandros Damalas[5], Shiri Ben Zvi[6], Aya Friedberg[6], Netta Eitan[6], Shachar Erez[6], Ada Yonath[3], Igor Ulitsky[4], Moshe Oren[1]

**ER stress underlies numerous severe pathologies. We have metabolically perturbed normal fibroblasts to study the biological roles of microRNAs (miRs) under mild and extended ER stress. We now report that miR-4488 quenches inflammation-associated gene expression in such metabolically perturbed cells. Remarkably, generation of miR-4488 is Drosha-independent. Furthermore, we define miR-4488 as a noncanonical miRNA derived from the expansion segment ES7L of the 28S ribosomal RNA. Moreover, its generation involves the autophagy–lysosome route and is inhibited when this pathway is blocked, thus unveiling an anti-inflammatory role for ribosomal RNA and lysosomes, engaged at the onset of stress. Mechanistically, miR-4488 suppresses the expression of NFKB2 and RELB, whose mRNAs specifically associate with miR-4488 exclusively upon stress. This selectivity suggests that miR-4488 may bear promise for treating mild ER stress–associated diseases.**

## Introduction

Metabolic stress is often associated with ER stress, which can contribute to a wide spectrum of disorders, including obesity, diabetes, liver diseases, Alzheimer's disease, and cancer, to name just a few (Harding & Ron, 2002; Malhi & Kaufman, 2011; Wang & Kaufman, 2016; Urra et al, 2016; Oakes, 2020; Chen & Cubillos-Ruiz, 2021; Ajoolabady et al, 2022a, 2022b). ER stress is ignited by an overload of misfolded ER-resident proteins, thus disrupting homeostasis. ER stress can be resolved by increasing the capacity of the ER to handle misfolded proteins and, in parallel, by decreasing the production of such proteins (Walter & Ron, 2011), by routes collectively referred to as the unfolded protein response (UPR) and ER-associated degradation of misfolded proteins (ERAD) (Mori, 2009; Walter & Ron, 2011; Gardner et al, 2013; Sun & Brodsky, 2019; Lemberg & Strisovsky, 2021). The UPR is governed by signaling pathways emanating from three ER membrane-embedded receptors—PERK, IRE1a (encoded by ERN1), and ATF6 (Mori, 2009; Walter & Ron, 2011; Wang & Kaufman, 2016). PERK leads to selective up-regulation of the translation of ATF4, a transcription factor with a pivotal role in the UPR (Harding et al, 2003; Lu et al, 2004; Baird & Wek, 2012; Arensdorf et al, 2013; Hetz et al, 2015; Pakos-Zebrucka et al, 2016; Renz et al, 2020). We have previously established an experimental system based on human primary fibroblasts undergoing mild and progressive ER stress upon starvation for glucose and serum (GS starvation) (Michael et al, 2023), and showed that up-regulated ATF4 and IRE1a drive the UPR and a cytokine and chemokine response (Michael et al, 2023). Furthermore, this system also incorporates changes in the expression patterns of several microRNAs (miRNAs).

Canonical miRNAs are ~22-nucleotide-long RNAs that direct posttranscriptional repression of mRNA targets, with profound roles in development and homeostasis (Pasquinelli & Ruvkun, 2002; Ambros, 2011; Pauli et al, 2011). Their deregulation has been linked to many disorders (Esteller, 2011). The canonical pathway that gives rise to mature miRNAs starts typically with transcription by RNA polymerase II (Pol II), leading to the generation of a primary transcript that undergoes capping and can occasionally also undergo polyadenylation (Bracht et al, 2004; Lee et al, 2004; Rodriguez et al, 2004), although some notable exceptions have been reported (Bartel, 2018). The canonical primary transcript (pri-miRNA) contains one or several stem–loop structures that are processed by the microprocessor complex, which contains the Drosha RNase III endonuclease, generating a ~60–70 nucleotide precursor miRNA (pre-miRNA) (Lee et al, 2003; Nicholson, 2014). This precursor undergoes nuclear-to-cytoplasmic export and is then processed by the RNase III enzyme Dicer to generate double-stranded miRNA duplexes (Macrae et al, 2006; MacRae et al, 2007). The miRNA duplex is loaded into a silencing complex that contains an Argonaute protein, and eventually, the mature single-stranded miRNA is retained by the Argonaute to target substrate mRNAs (Matranga et al, 2005; Rand et al, 2005; Gebert & MacRae, 2019).

Here, we report that miR-4488 is up-regulated at the onset of starvation stress. Unexpectedly, it is produced via a noncanonical,

---

[1]Department of Molecular Cell Biology, The Weizmann Institute of Science, Rehovot, Israel [2]The Weizmann School of Science, The Weizmann Institute of Science, Rehovot, Israel [3]Department of Chemical and Structural Biology, The Weizmann Institute of Science, Rehovot, Israel [4]Department of Immunology and Regenerative Biology and Department of Molecular Neuroscience, The Weizmann Institute of Science, Rehovot, Israel [5]Department of Biology, Faculty of Medicine, University of Thessaly, Biopolis, Larissa, Greece [6]Robert H. Smith Faculty of Agriculture, Food and Environment, Hebrew University of Jerusalem, Rehovot, Israel

Correspondence: d.michael@weizmann.ac.il, dan.michael22@gmail.com; moshe.oren@weizmann.ac.il

Drosha-independent route, albeit in a Dicer-dependent manner. Furthermore, we provide evidence that the mature miR-4488 sequence originates in the 28S ribosomal RNA (rRNA). Surprisingly, it can be produced through the macroautophagy–lysosome pathway before being processed by Dicer. After an initial boost in its production in GS-starved fibroblasts, the endogenous miR-4488 is down-regulated 48 h after the onset of starvation, suggesting a role of this miRNA in restraining spurious and premature responses to metabolic stress. Moreover, its overexpression can suppress the ATF4 and ERN1 ER stress–associated proinflammatory phenotype, robustly taming ER stress and down-regulating the expression of many chemokines and cytokines. We identify the key inflammatory drivers NFKB2 and RELB as direct miR-4488 targets.

Overall, our findings reinforce the message that lysosomes are not just recycling bins. Moreover, they imply that lysosomes can assume an anti-inflammatory role by becoming a source of a ribosome-derived miRNA that enables a unique mode of metabolic regulation, delaying premature ER stress–associated responses to optimize cellular metabolic adaptation. Furthermore, as the biological effects of miR-4488 are largely dependent on the specific stress conditions, we propose that this miRNA may be used therapeutically to ameliorate ER stress–associated pathologies, leaving the nonstressed cells in the body unaffected.

## Results

### miR-4488 is produced in a Drosha-independent but Dicer-dependent manner

In metabolically challenged human foreskin fibroblasts (HFFs), where ATF4 and ERN1 have a pivotal role in ER stress–associated proinflammatory responses, we observed changes in the expression of several microRNAs 48 h after GS starvation (Michael et al, 2023). One of those was miR-4734, whose levels were progressively reduced as the stress gradually intensified, and which we recently characterized (Michael et al, 2023). Based on our previous microarray analysis (Michael et al, 2023), we next focused on another miRNA, miR-4488, whose levels, similar to miR-4734, changed upon 48 h of starvation. First, we monitored the levels of miR-4488 at different times after the onset of starvation. As seen in Fig 1A although the levels of miR-21-5p, serving as a control, were unaffected by starvation, miR-4488 levels increased by 4 h of GS starvation but declined progressively at 24 and 48 h (Fig 1B). To obtain an estimate of the number of miR-4488 copies per cell, we performed an RT-absolute quantification PCR as described in the Materials and Methods section. Despite some variation among biological repeats, the values obtained indicated that miR-4488 is relatively abundant (4,229, 1,350, 2,090 copies per cell, with an average of 2,556 copies and a SD of 1,495). Given that miR-4488 has not been studied extensively, we then proceeded to determine its mode of production. Specifically, we employed 293T Drosha knockout cells (Park et al, 2018) and HCT116 Dicer[ex5] hypomorph cells (Cummins et al, 2006) to investigate whether production of miR-4488 depends on Drosha and/or Dicer. As expected, the control miRNA, miR-21-5p, required Drosha for its production and

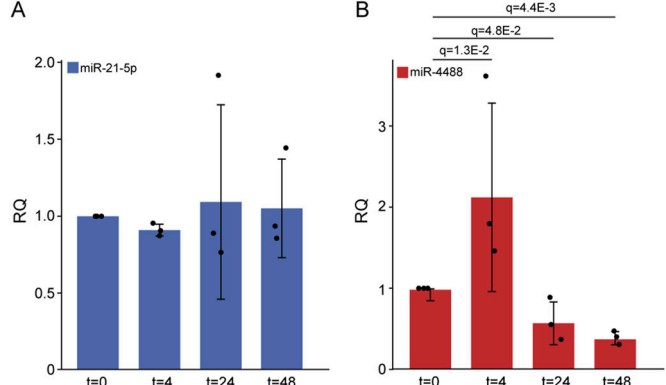

**Figure 1. miR-4488 levels increase at the onset of starvation but decrease upon prolonged starvation.**
HFFs were subjected to GS starvation for the indicated periods (in hours). **(A, B)** Expression levels of (A) miR-21-5p and (B) miR-4488 were determined by RT–qPCR. Relative quantity (RQ) was calculated by normalizing each value to the corresponding expression of the miRNA in nonstarved cells. Data are from three biological replicates. ANOVA was used to calculate the significance of the changes at different time points against time = 0. Delta Ct values of the replicates were used as the input for ANOVA, and the replicates were added as a random factor. RQ values with standard deviations and false discovery rates (q-values) are shown.

its levels were dramatically diminished in the Drosha-deficient cells (Fig 2A). Surprisingly, miR-4488 levels not only did not decrease but actually even increased in the Drosha knockout cells (Fig 2A), implying that its production is Drosha-independent and raising the possibility that the increase is due to some stress inherent to the Drosha knockout cells. In contrast, the generation of miR-4488 in Dicer[ex5] hypomorph cells was compromised to a similar extent as that of miR-21-5p (Fig 2B). Hence, miR-4488 appears to be produced via a Dicer-dependent but Drosha-independent route. Finally, to probe for the association of the endogenous miR-4488 with Argonaute proteins, we performed an Ago-APP (Ago protein Affinity Purification by Peptides) assay with a short TNRC6B-derived peptide (T6B), which efficiently interacts with Argonaute family proteins (Hauptmann et al, 2015; Hauptmann & Meister, 2017). This assay is designed to pull down the Argonaute-associated microRNAs. The extent of pulldown by the WT peptide is compared with that obtained with a mutant peptide, which is incapable of binding Argonautes. As seen in Fig 2C, miR-4488 was enriched in the WT peptide pulldown as compared to the mutant peptide, implying that miR-4488 can associate with Argonaute proteins.

### miR-4488 can be generated from ribosomal RNA

The noncanonical generation of miR-4488 raised the possibility that this miRNA might be generated by a noncanonical pathway. This microRNA is annotated by miRBase to originate from the human genome at chr11: 61508596–61508657 (plus strand, GRCh38 assembly) (Kozomara & Griffiths-Jones, 2014). Surprisingly, however, alignment to the human genome revealed the sequence of the mature miR-4488 in 219 different 28S ribosomal RNA (rRNA) loci (Table S1). This is in line with an earlier suggestion that miR-4488 might be potentially derived from mature 28S rRNA

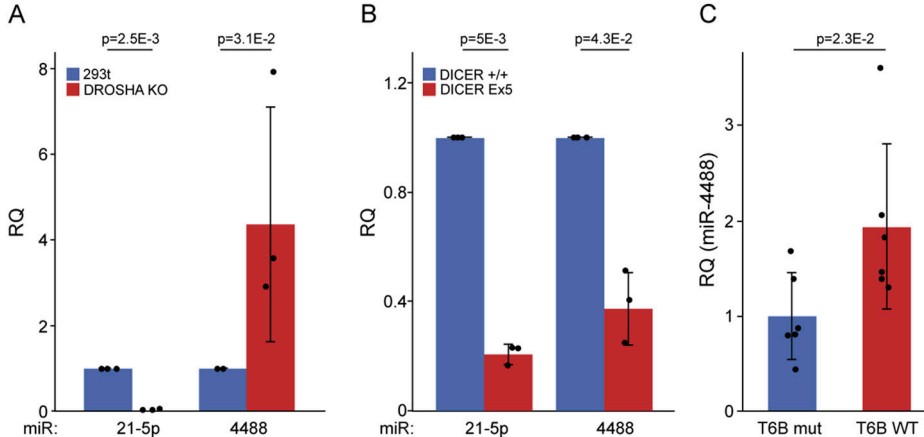

**Figure 2. miR-4488 is produced in a Drosha-independent but Dicer-dependent manner.**
**(A)** Comparison of the relative levels of the indicated miRNAs in control 293T cells (taken as 1.0) and their *Drosha* knockout derivatives, determined by RT–qPCR. **(B)** Comparison of the relative levels of the indicated mature miRNAs in control (DCR+/+) HCT116 cells (taken as 1.0) and their derivatives expressing homozygously a mutant hypomorphic allele of Dicer (DCR Ex5), determined by RT–qPCR. RQ = relative quantity. Data are from three biological replicates. Relative quantity (RQ) was calculated by normalizing the relative expression of the miRNA in knockout or knock-in cells to its expression in control cells. Delta Ct values of the replicates were used as the input for ANOVA, and the replicates were added as a random factor. Standard deviations and *P*-values are shown. **(C)** Enrichment of miR-4488 in Ago-APP pulldown employing the WT TNRC6B-derived peptide (T6B) as a bait, relative to the effect of the mutant peptide incapable of specifically binding Argonautes. Values in the pulldown and input were normalized to the nonspecific binding of SNORD44. Data analysis was performed similar to the analysis described in the previous panels, except that the data were extracted from six biological repeats.

(Yoshikawa & Fujii, 2016). Within the mature ribosome, the miR-4488 sequence maps to the human expansion segment ES7L of the 28S rRNA (see below). Interestingly, expansion segments are ribosome regions that have expanded during evolution, and their function in modulating translation dynamics is only partially understood (Hariharan et al, 2023; Rauscher & Polacek, 2024). ES7L is one of the largest known expansion segments in eukaryotes (Fig S1), and it comprises ~247 nt (from ~454 to 701 in the 28S rRNA). The 3D structure of ES7L is highlighted as a red ribbon in Fig 3A. Region 518–642nt, in dotted blue lines, was not modeled in any human ribosome cryo-EM structures to date, most likely because of the large flexibility of this surface rRNA helix, which results in poor EM map resolution in this region. As shown in Fig 3B, miR-4488 can potentially be derived from the nonmodeled part in this structure and matches the sequence at 586–603nt. Moreover, further analysis of the corresponding region of the 28S rRNA identified a stem–loop structure that resembles a pre-miRNA, with incomplete base pairing of the stem part (Fig 3C).

If miR-4488 can indeed originate from the processing of rRNA, we would expect at least some endogenous copies of this miRNA to possess, at their 3′ end, unique nucleotide signatures that are only present in the sequence of the 28S rRNA but not in that of the genomic site in chr11 harboring the annotated miR-4488. We therefore performed a targeted deep sequencing analysis, using a miR-4488 primer to derive the second-strand DNA, followed by PCR amplification (see the Materials and Methods section). This analysis yielded a total of 17,716 unique molecular identifiers (filtered UMIs) (Fig 3D and Table S2). Of those, about 23.5% were 18 nucleotides or shorter and thus could be derived either from the annotated genomic miR-4488 locus or from rRNA. Importantly, the remainder of the reads, 19 nucleotides or longer, were found to be derived almost exclusively from 28S rRNA sequences, whereas only 0.83% of the total UMI reads could be assigned to the genomically annotated miR-4488 locus. All the different variants of the sequenced miR-4488 matched perfectly the sequence of the 28S RNA, irrespective of their length (Fig S2). Hence, the great majority of mature miR-4488 molecules originate from rRNA-encoding loci, perhaps through processing of 28S rRNA.

## Production of miR-4488 requires autophagy and lysosomal function

If miR-4488 is indeed produced from 28S rRNA, it may be produced either from the precursor rRNA (before ribosome assembly) or from mature 28S rRNA embedded in ribosomes. In the latter case, one might consider several alternative options for the production of pre-miR-4488 from 28S rRNA. One of those is nonfunctional ribosomal RNA decay, a canonical turnover pathway that degrades rRNA from ribosomes that are defective in translation (LaRiviere et al, 2006; Cole et al, 2009; Fujii et al, 2009). This pathway was characterized primarily in yeast, and it is unclear whether it is faithfully represented in mammals. Another option involves a distinct autophagy pathway in which RNAs, as well as DNAs, are directly imported into lysosomes for degradation; in the case of RNA, this pathway is referred to as "RNautophagy." This pathway requires two lysosomal membrane proteins, LAMP2C and SIDT2, to directly take up nucleic acids for lysosomal degradation (Fujiwara et al, 2013; Aizawa et al, 2016, 2017; Hase et al, 2020). We therefore knocked down SIDT2 and monitored miR-4488 levels. Despite the efficient knockdown of SIDT2 mRNA (Fig S3A), no significant down-regulation of miR-4488 was observed (Fig S3B), arguing against the possibility that miR-4488 is generated via RNautophagy. Yet, it remained possible that miR-4488 is generated from 28S rRNA by a different autophagy-dependent process. In particular, autophagy-dependent lysosomal degradation of ribosomes is known to occur under various stress conditions, including starvation (Kraft et al, 2008; An & Harper, 2018; Wyant et al, 2018; López et al, 2023). Furthermore, mTOR inhibition, which mimics many aspects of starvation, also leads to ribosome autophagy, which requires the class III PI3K VPS34 protein working in conjunction with BECLIN1 (An & Harper, 2018).

To test the possible involvement of the autophagosome-to-lysosome pathway in the generation of miR-4488, we used three inhibitors and one activator of this pathway (Whitmarsh-Everiss & Laraia, 2021). SAR405 is an inhibitor of autophagy initiation, which interferes with autophagosome formation at the nucleation step,

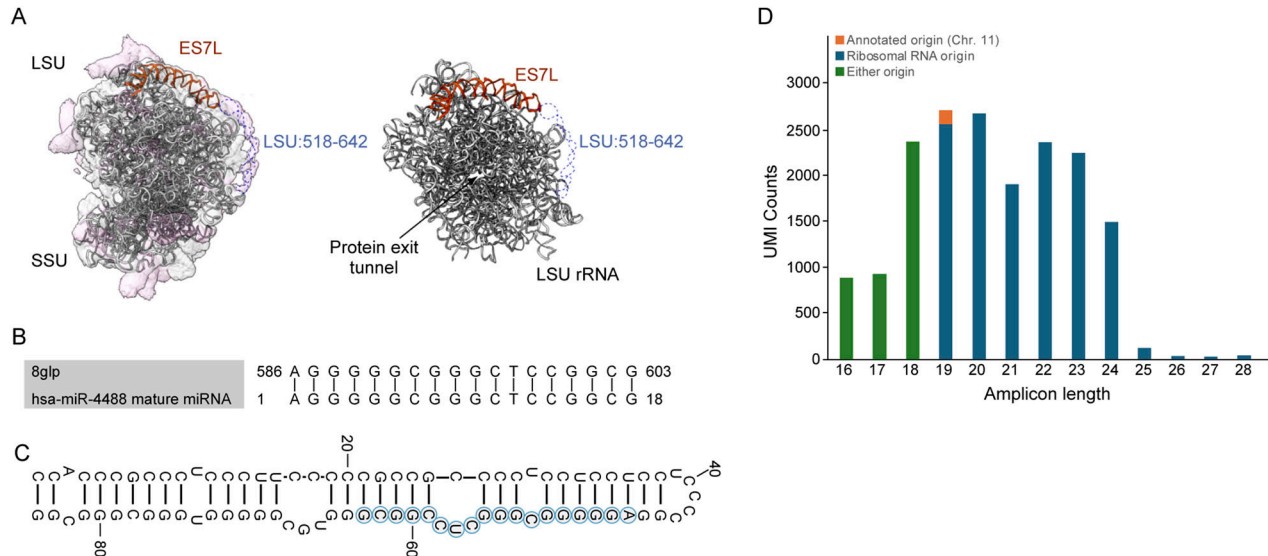

**Figure 3. miR-4488 can be derived from 28S ribosomal RNA.**
**(A)** Location of ES7L in the 1.67 Å cryo-EM structure of human ribosomes. The atomic coordinates and EM map were derived from PDB-8GLP and EMD-40205. The modeled region of ES7L is highlighted in a red ribbon, and the unmodeled region is indicated in blue dotted lines. The EM map and atomic coordinates are shown as surface and ribbon. **(B)** Predicted origin of miR-4488 within human 28S rRNA (nt 586–603), not modeled in any cryo-EM structure available to date. **(C)** Secondary structure prediction of pre-miR-4488 using RNAfold (http://rna.tbi.univie.ac.at/). The mature miR-4488 sequence is highlighted. **(D)** Determination of the origins of endogenous miR-4488 by sequencing of library amplicons derived from a miR-4488 core primer used for cDNA generation. The library was generated as described in the Materials and Methods section. The resulting unique sequence copies were classified according to their length and the compatibility of their sequence with either 28S rRNA (blue) or the genomic DNA locus predicted to give rise to miR-4488 (orange), or either origin if it cannot be decided (green).

potently inhibiting VPS34 (Ohashi, 2021). Bafilomycin A1 is a potent inhibitor of the multi-subunit v-ATPase proton pump, which is responsible for maintaining low lysosomal pH and whose inhibition can block the fusion of lysosomes with upstream autophagosomes. Chloroquine is a lysosomotropic agent that blocks the ability of the lysosome to degrade macromolecules. As shown in Fig 4A–C, all three inhibitors of the autophagosome-to-lysosome pathway substantially suppressed the increase of miR-4488 at 4 h post-GS starvation. The relative increase in miR-4488 levels after the 4-h starvation treatment, assessed by combining the data in Figs 1B and 4A–C, is presented in Fig S4. We next extended the analysis using Torin-1, an inhibitor of TORC1 and TORC2, which activates autophagy (Liu et al, 2010). Remarkably, Torin-1 significantly enhanced the production of miR-4488 (Fig 4D). Together, our findings suggest that miR-4488 may be derived from 28S rRNA that reaches the lysosome via autophagy. After this initial processing in the lysosome, the production of mature miR-4488 is presumably completed in the cytoplasm with the help of Dicer. To further support this model, we monitored the precursor of miR-4488 (pre-miR-4488). To this end, we used a primer ending just before the first nucleotide of the mature miR-4488 sequence (Fig S5A) in order to quantify the relative amounts of pre-miR-4488 in different subcellular compartments. As shown in Fig S5B, this precursor was found in the lysosome-containing fraction and in the cytosol, consistent with a mechanism wherein it is generated in the lysosome and then transported into the cytoplasm for further processing into mature miR-4488.

## miR-4488 restricts ER stress–associated proinflammatory responses

As shown in Fig 1, miR-4488 levels rise early during GS starvation but decline progressively by 48 h. This occurs concurrently with the up-regulation of ER stress–associated genes, as well as cytokines and chemokines (Michael et al, 2023), raising the possibility that miR-4488 restricts the early induction of stress-associated responses, whereas its later decline unleashes these responses. To test this hypothesis, as well as to identify potential miR-4488 targets, we pretransfected HFF cells with exogenous miR-4488 and monitored its impact on global gene expression after 48 h of GS starvation. Altogether, we detected 4,232 differentially expressed informative genes. A heatmap summarizing this analysis is shown in Fig 5A. Of particular note was cluster 4, comprising genes that were up-regulated upon starvation (t = 48 miR-CTRL versus t = 0 miR-CTRL), but whose up-regulation was markedly quenched by miR-4488 overexpression (t = 48 miR-4488 versus t = 48 miR-CTRL). We then focused on the entire list of miR-4488-down-regulated genes irrespective of their cluster origin, ending up with a set of 409 genes that were significantly down-regulated (t = 48 miR-4488 versus t = 48 miR-CTRL, fold change of –1.5 or less and q-value of 0.05 or less). Ingenuity Pathway Analysis revealed that this set was strongly enriched with genes encoding proteins implicated in proinflammatory response and cancer (Fig 5B), including numerous cytokines and chemokines and other proinflammatory proteins (Table S3 and Fig 5C). Thus, miR-4488 negatively regulates the expression of proinflammatory genes.

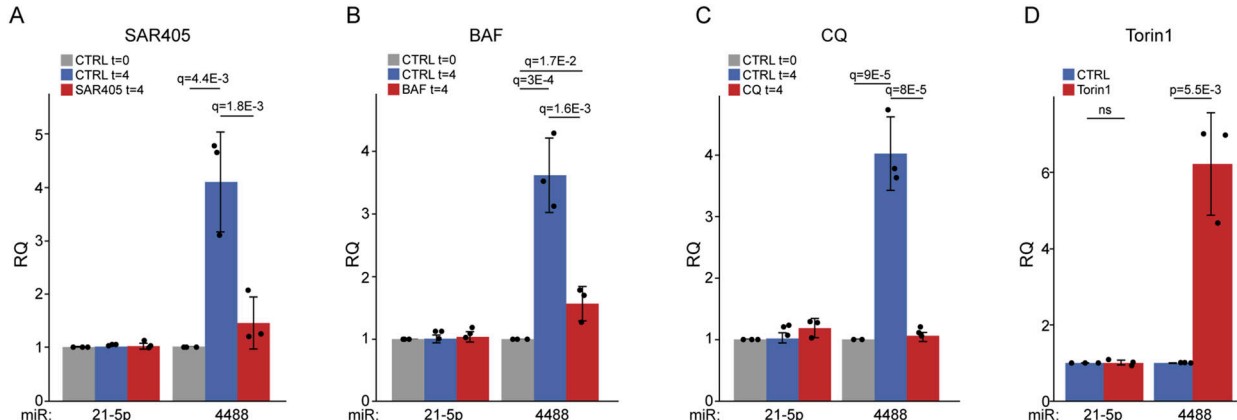

**Figure 4.  miR-4488 is produced via the autophagosome-to-lysosome pathway.**
**(A, B, C, D)** HFF cells were kept in basal conditions (t = 0) or starved (t = 4) in the absence (CTRL) or presence of either (A) 0.3 $\mu$M SAR405, (B) 50 nM bafilomycin A (BAF), (C) 20 $\mu$M chloroquine (CQ), or (D) cells were not starved but instead treated with 200 nM Torin-1 (depicting results only at t = 4). The expression of miR-21-5p and miR-4488 was quantified by RT–qPCR. Relative quantity (RQ) was calculated by normalizing the relative expression of the miRNA in starved cells to its expression in control cells (t = 0). For statistical analysis procedures, see legend to Fig 1.

To monitor more rigorously the effect of miR-4488 over-expression on genes associated with the proinflammatory response (*IL8* [*CXCL8*], *CXCL1*, *IL1A*, and *IL1B*) and with the ER stress response (*ERO1LB*, *DNAJB9*, *HSPA5*, *DDIT3*, *ERN1*, and *ATF4*) in the course of increasing metabolic challenge (Michael et al, 2023), we performed RT–qPCR analysis. First, we monitored gene expression after starvation using as a reference miR-CTRL–transfected cells kept at basal conditions (t = 0). Upon intensified stress, and in the presence of miR-CTRL, the expression of both gene groups increased (Fig 6A and Supplemental Data 1), as expected (Michael et al, 2023). As seen in Fig 6B, over the same reference, in non-starved cells and at 4 h of starvation, the expression of those genes was not affected by miR-4488 overexpression. However, the expression of this entire set of genes was down-regulated by miR-4488 at 24 h of starvation, whereas the proinflammatory genes were also down-regulated at 48 h, often even more strongly than at 24 h (Fig 6B). These results underscore a gene-selective impact of miR-4488 overexpression under different conditions of stress, suggesting a conditional effect of exogenous miR-4488.

To ascertain that the endogenous miR-4488 also contributes to the mitigation of the stress response, we employed a miR-4488 hairpin inhibitor. As seen in Fig 6C, using cells pretransfected with the HP NEG-CTRL under resting conditions as a reference, the expression of ER stress and proinflammatory genes increased in cells transfected with the same control inhibitor, in particular at 48 h. Remarkably, as seen in Fig 6D, the HP-INHIB-4488 inhibitor led to up-regulation of proinflammatory genes over the reference control, specifically at the basal (nonstarved) conditions and at 4 and 24 h of starvation, whereas the expression of ER stress–associated genes was up-regulated mostly at 24 h. Of note, the effect of the miR-4488 inhibitor was lost at 48 h, corresponding to a time point when the endogenous miR-4488 levels become very low and may be insufficient to restrain the expression of its target genes. Interestingly, we have previously observed that the inhibition of miR-4734 impacted gene expression only at 24 h of starvation (Michael et al, 2023). Therefore, miR-4488, unlike miR-

4734, may restrict the proinflammatory response under low stress, including stress originating from basal cell culture conditions, whereas miR-4734 may operate only under relatively stronger stress.

ATF4 is a key component of the ER stress response. However, it is regulated primarily at the protein rather than the RNA level. We therefore monitored the effect of miR-4488 overexpression on ATF4 protein levels without and with metabolic stress. As seen in Fig 7A and B, ATF4 levels were significantly reduced by miR-4488 in both nonstarved and starved cells, particularly at 4 and 24 h after the induction of stress. The effect in nonstarved cells resonated with the notion that miR-4488 may restrict the stress associated with tissue culture conditions, as observed also with the miR-4488 hairpin inhibitor (Fig 6B). Of note, IRE1a (encoded by ERN1) and ATF4 cooperate in up-regulation of proinflammatory genes in our experimental model (Puschel et al, 2020; Michael et al, 2023). We therefore next monitored the impact of miR-4488 on IRE1a protein levels. As seen in Fig 7A and C, IRE1a protein levels were decreased by miR-4488 only at late phases of starvation (24 and 48 h), coinciding with the down-regulation of the proinflammatory genes. Thus, miR-4488 conditionally restricts ATF4- and IRE1a-dependent proinflammatory responses. Interestingly, the reduced expression of miR-4488 was observed in Behcet's disease patients, who are prone to systemic recurrent inflammation (Woo et al, 2016). In addition, miR-4488 was underrepresented in inflamed venous endothelial cells, and miR-4488 mimic inhibited the accumulation of inflammatory proteins under conditions of arterial laminar shear stress (Fang et al, 2021). These observations are consistent with a role of miR-4488 in restraining inflammation.

## miR-4488 conditionally associates with *NFκB2* and *RELB* mRNA

We next wished to identify genes whose transcripts physically interact with miR-4488. Given the role of the IRE1a-NF-κB axis in proinflammatory responses (Kaneko et al, 2003; Schmitz et al,

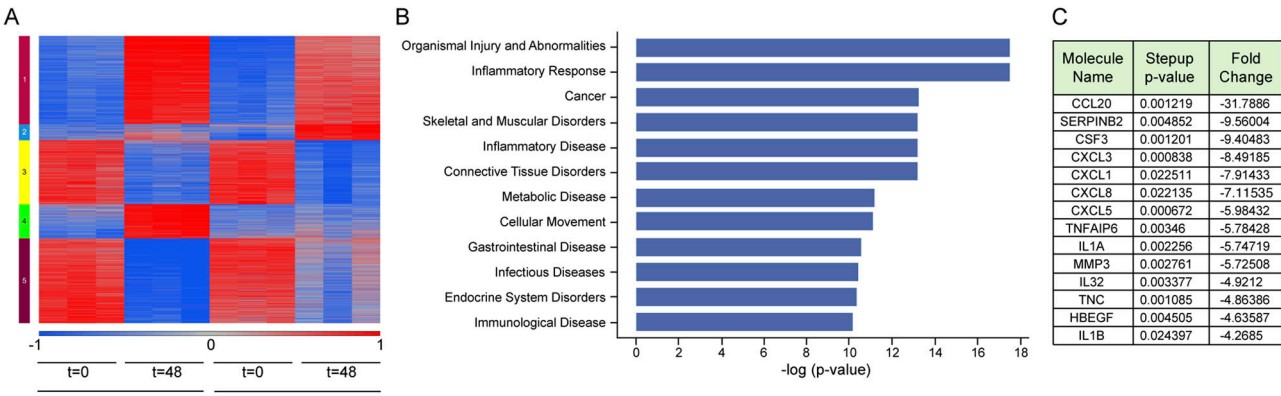

**Figure 5. miR-4488 restricts the expression of proinflammatory cytokines and chemokines.**
Global effects of miR-4488 on mRNA expression in starved cells. **(A)** Cells were transiently transfected with 10 nM miR-4488 mimic or control miRNA (miR-CTRL). 48 h later, the cells were either subjected to GS starvation for 48 h or not starved, followed by Affymetrix expression array analysis. The heatmap shows the partitioning of differentially expressed genes characterized by an absolute fold change of at least two and a false discovery rate of 0.05 in at least one pairwise comparison. The 4,232 differentially expressed informative genes were divided into five clusters (see the Materials and Methods section). **(B)** Functional analysis of the 409 genes significantly down-regulated by miR-4488 in starved cells (t = 48 miR-4488 versus t = 48 miR-CTRL, fold change of −1.5 or less and q-value of 0.05 or less, using Ingenuity Pathway Analysis). **(C)** List of the genes encoding secreted factors that were most strongly down-regulated by miR-4488 in starved cells (t = 48).

2018), we specifically focused on NF-κB family members. In our global gene expression assay, the levels of *NFKB2* mRNA, and to a lesser extent of *NFKB1* mRNA, were down-regulated by miR-4488 at 48 h (1.7- and 1.3-fold reduction, respectively). Seed sequences at nucleotide positions 2–8 or 2–7 (counting from the 5′-end) of a miRNA are the primary determinants of miRNA-based mRNA targeting (Bartel, 2009). *NFKB2* mRNA has one 7mer miR-4488 seed site in the coding sequence and two 6mer seed sites in its 3′UTR. Of note, binding of miRNAs within coding sequences has already been implicated in posttranscriptional regulation (Forman et al, 2008; Fang & Rajewsky, 2011; Reczko et al, 2012). *NFKB1* mRNA, on the other hand, has just one 6mer candidate seed binding site, positioned in the coding sequence. An additional NF-κB family member is RELB, which is a potent transcriptional activator upon association with p50 (encoded by *NFKB1*) or p52 (encoded by *NFKB2*) (Bours et al, 1992; Ryseck et al, 1992; Baud & Collares, 2016; Rodriguez et al, 2024), whose expression is significantly up-regulated by GS starvation (Michael et al, 2023). *RELB* mRNA has one 7mer and six 6mer seed sites, all in the coding sequence. As shown in Fig 8A, all three genes were indeed up-regulated upon stress and were conditionally down-regulated by miR-4488. To determine whether any of these transcripts are direct targets of miR-4488, we performed pulldown (PD) analysis on cells transfected with biotinylated miR-4488, followed by RT–qPCR quantification. As seen in Fig 8B and C, only *NFKB2* and *RELB* mRNAs were pulled down with miR-4488. In addition, we performed luciferase-based analysis of the effect of miR-4488. Fig S6A shows the candidate binding sites of miR-4488 in the 3′UTR of *NFKB2* mRNA. As seen in Fig S6B, miR-4488 exerted a modest inhibitory effect on luciferase activity. This inhibitory effect was very partly attenuated upon mutation of the predicted miR-4488 binding site, suggesting that miR-4488 likely acts through additional sites or other mechanisms. Importantly, the associated mRNAs in the pulldown assays suggested a conditional response, observed only upon stress but not under basal conditions. We thus propose that by

directly down-regulating key NF-κB components in the IRE1a-NF-κB pathway in a context-dependent manner, miR-4488 can restrict the undesirable expression of proinflammatory genes.

## Discussion

This study describes a novel pathway wherein ribosomal RNA is processed by the lysosome to give rise to a microRNA that restricts stress-induced inflammatory responses, at least in part by targeting components of the NF-κB family of regulators (Kaneko et al, 2003; Schmitz et al, 2018; Puschel et al, 2020). Of note, it is increasingly being recognized that small RNA fragments, and particularly ribosomal RNA–derived fragments (rRFs), have numerous attributes that underscore their biological significance (Lambert et al, 2019; Lingyu & Andrey, 2020; Rosace et al, 2020; Cherlin et al, 2024). Thus, rRFs display precise cleavage patterns and distinct size distributions, rendering it unlikely that they are generated by random degradation (Li et al, 2012; Chen et al, 2017). Our observation that most endogenous miR-4488 products are about 20–24nt long (Fig 3D) is consistent with this notion. Interestingly, rRFs can be expressed as a result of additional homeostatic perturbations, as shown in *Saccharomyces cerevisiae* in which rRFs are mainly derived from the 25S rRNA under oxidative stress conditions (Thompson et al, 2008; Thompson & Parker, 2009). We now show that production of 28S-derived miR-4488 is up-regulated at the onset of metabolic stress in normal human cells (Figs 1, 3, and 4). Similar to what we report for miR-4488, some rRFs are produced in a Drosha-independent but Dicer-dependent manner, whereas others are produced in a Dicer-independent manner (Rosace et al, 2020). Of note, our conclusion that miR-4488 is produced in a Dicer-dependent manner is based on results obtained in cancer-derived HCT116 cells. Although this system is advantageous given that all the Dicer molecules are equally

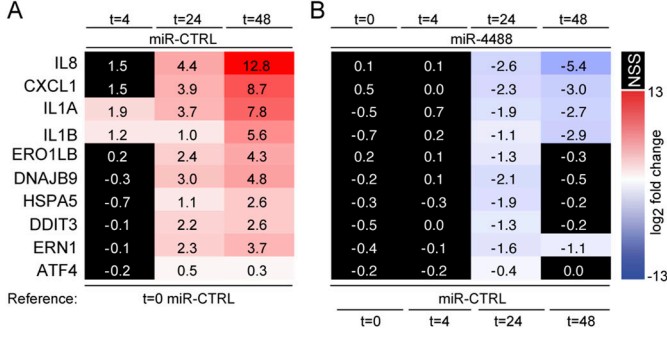

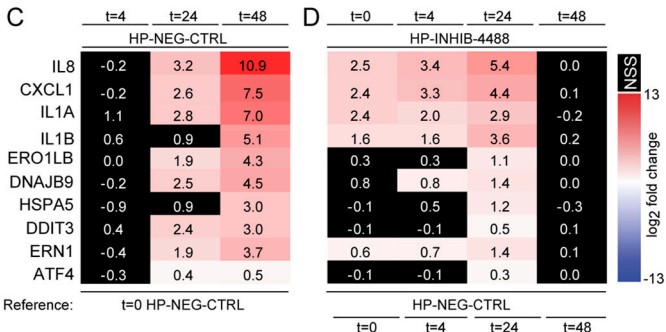

**Figure 6. miR-4488 attenuates the induction of inflammation and ER stress–associated genes.**

**(A, B)** HFF cells were transfected with either 10 mM control miRNA (miR-CTRL) or 10 nM miR-4488 mimic and harvested either 48 h later (t = 0) or after additional GS starvation for the indicated time periods (in hours). Relative gene expression levels were determined by RT–qPCR. Relative quantity (RQ) was calculated by normalizing the relative expression of the miRNA in starved cells to its expression in cells transfected with miR-CTRL. Numbers represent average log$_2$ fold change obtained from three biological replicates. ANOVA was used to calculate the significance of the changes at the different time points. Statistical significance values are listed under Supporting Statistical Analysis. Nonstatistically significant values are colored black. **(C, D)** Cells were transfected with either 20 nM hairpin negative control (HP-NEG-CTRL) or 20 nM miR-4488 inhibitor (HP-INHIB-4488), and 48 h later, they were either harvested (t = 0) or subjected to GS starvation for the indicated time periods. Relative gene expression and statistical significance were similarly determined as in (A); values are shown in Supplemental Data 1.

compromised in their activity, we cannot rule out that in other cell types additional modes of generation of mature miR-4488 may also exist. Additional nucleases, such as XRN1, have also been shown to participate in generating rRFs, whereas in many other cases, the responsible nucleases remain unknown (Rosace et al, 2020). Mechanistically, as reported here for miR-4488, some other rRFs have been shown to associate with Argonaute proteins, promoting silencing of gene expression (Li et al, 2012; Wei et al, 2013; Chak et al, 2015; Rosace et al, 2020). Furthermore, as sporadically shown for some rRFs (Son et al, 2013; Wei et al, 2013), miR-4488 is able to fine-tune cellular processes. Importantly, our work reveals a novel pathway for the generation of a functional rRF, involving autophagy and lysosomal activity, and suggests an anti-inflammatory role for lysosomes.

To some extent, the anti-inflammatory transcriptional effects of miR-4488 resemble those of miR-4734 (Michael et al, 2023). Similar to some other miRNAs (Mendell & Olson, 2012), both can be viewed as modulators of stress signals. However, the different kinetics of their induction and the distinct consequences of inhibition of the endogenous miRNAs suggest that miR-4488 may tame the proinflammatory response already at low levels of stress, whereas miR-4734 restricts the proinflammatory response only at a higher threshold. We propose that both miRNAs have the capacity to buffer against stochastic fluctuations in gene expression, avoiding spurious, unwanted proinflammatory responses (Vidigal & Ventura, 2015). Furthermore, they may enforce a time window within which transient stress is prevented from inducing the production of cytokines, whose unwelcome presence may have deleterious consequences. We propose that the transient action of miR-4488 enables cells to distinguish between relatively harmless time-limited stress conditions and other scenarios that involve progressive and extended stress, the latter necessitating an effective proinflammatory response. Mechanistically, miR-4488 targets both *NFKB2* and *RELB* mRNAs, and these well-studied transcriptional modulators can dimerize and activate proinflammatory gene expression (Sun, 2011; Baud & Collares, 2016; Mockenhaupt et al, 2021; Rodriguez et al, 2024). Our results are

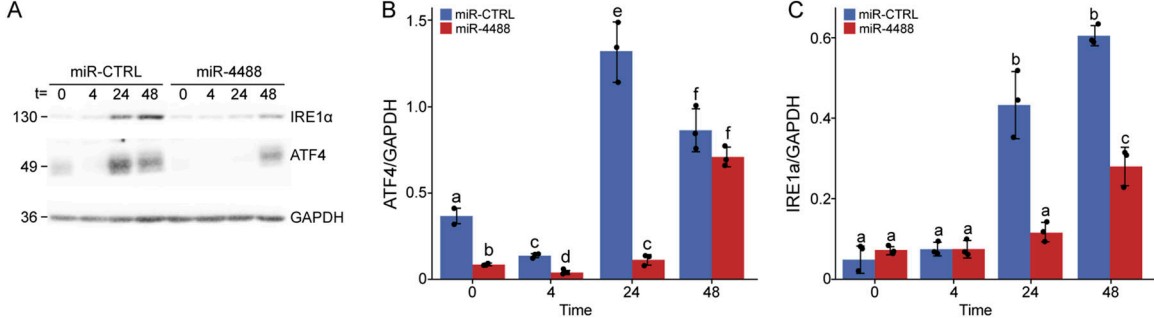

**Figure 7. miR-4488 down-regulates ATF4 and IRE1α protein levels.**

**(A)** HFF cells were transfected with either 10 mM miR-CTRL or 10 mM miR-4488 mimic and were either harvested 48 h later (t = 0) or subjected to GS starvation for the indicated time periods before harvesting. ATF4 and IRE1α proteins were assessed by Western blot analysis. GAPDH served as a loading control. **(B, C)** Quantification of ATF4 and IRE1α, from three biological repeats. Significant differences between the treatments were determined by ANOVA. Treatments marked with different letters are significantly different ($P < 0.05$), whereas those that are marked by the same letter are not statistically significant. SDs are shown. Source data are available for this figure.

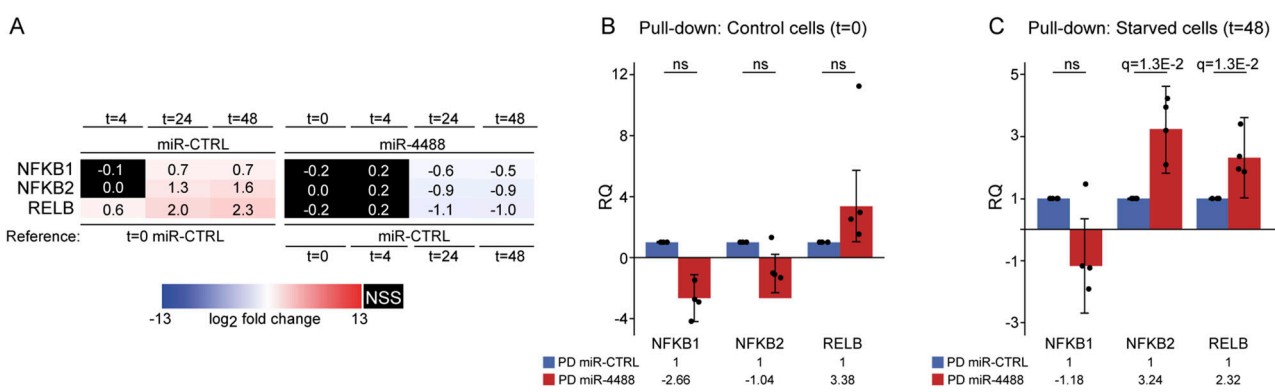

**Figure 8. *NFKB2* and *RELB* mRNAs conditionally associate with miR-4488.**
**(A)** HFFs were transfected with either miR-CTRL or miR-4488 mimic. 48 h later, the cells were either harvested (t = 0) or subjected to starvation for the indicated time periods. Relative expression of the indicated genes was determined by RT–qPCR (three biological repeats). Statistical significance values are listed under Supplemental Data 2. Nonstatistically significant (NSS) values are colored black. **(B, C)** HFFs were transfected with either biotinylated miR control or biotinylated miR-4488 (30 mM) and were either harvested after 48 h (t = 0) (B) or subjected to an additional 48 h of starvation (C). Pulldown (PD) analysis was performed using streptavidin-coated beads (see the Materials and Methods section), and the associated *NFKB1*, *NFKB2*, or *RELB* mRNAs were quantified by RT–qPCR (four biological replicates).

in agreement with other examples of a single miRNA inhibiting simultaneously multiple components of the same pathway to elicit a specific biological effect (Mendell & Olson, 2012). Importantly, although both *NFKB2* and *RELB* mRNAs are direct miR-4488 targets, this miRNA associates with them only upon starvation (Fig 8B and C).

Here, we not only provide a biological and molecular detailed context to miR-4488 function, but we also show that its actions stem from a pathway wherein lysosomes play a role in mitigating premature proinflammatory responses at the posttranscriptional level. This novel pathway reinforces the recent evidence that lysosomes do not function solely as recycling bins but are involved in numerous regulatory processes, such as mTORC1-mediated nutrient sensing, orchestrating transcriptional programs that control various metabolic and cellular events (Saftig & Klumperman, 2009; Settembre et al, 2013; Perera & Zoncu, 2016; Davidson & Vander Heiden, 2017; Trivedi et al, 2020; Eriksson & Öllinger, 2024).

Finally, along with the previously described miR-4734 (Michael et al, 2023), miR-4488 may constitute a potential toolkit to treat a variety of disorders characterized by ER stress and inflammatory responses. The therapeutic importance of modulating ER stress has already been demonstrated in several settings (Grandjean & Wiseman, 2020; Marciniak et al, 2021; Chen et al, 2023). In that regard, the fact that the effects of these miRNAs are, by and large, conditional and are mainly exerted under stress conditions may render them particularly attractive, because their use may spare normal, nonperturbed tissues from undesirable side effects.

# Materials and Methods

### Cell culture

Cultures were grown in medium supplemented with FBS (12657-029; Gibco) and kept in a 5% $CO_2$ incubator at 37°C. HFFs (AG14608)

were obtained from the NIA Repository, administered by the Coriell Institute for Medical Research, Camden, New Jersey. Cells were grown in Eagle's Minimum Essential Medium with Earle's salts and nonessential amino acids (M5650; Sigma-Aldrich) supplemented with 2.6 mM L-glutamine, ampicillin and streptomycin, and 15% FBS. For routine culturing, cells were split 1:3 every 3–4 d. Before a typical experiment, 135,000–140,000 cells were seeded in a 6-cm dish and cultured at a density of 37,000 cells per 12-well plate and grown for 44–46 h before transfection. RNA molecules used for transfection and the DharmaFECT 1 transfection reagent were obtained from GE Dharmacon. Unless otherwise specified, the following siRNAs (siGENOME; Dharmacon) were used at 15 nM final concentration: siGENOME Control-pool Non-targeting #2 (siCTRL), siRNA SMARTpool directed against SIDT2 (siSIDT2), and siRNAs tested individually, which are listed in Table S4. With the exception of miR-mRNA pulldown experiments, all miRNA mimics listed in Table S4 were applied at 10 nM final concentration. miRIDIAN microRNA Hairpin Negative Control #1 (HP-NEG-CTRL) and miRIDIAN microRNA hsa-miR-4488 Hairpin Inhibitor (HP-INHIB-4488) were applied at 20 nM final concentration. All transfection mixes were made in Opti-MEM reduced serum medium (31985-047; Gibco/Thermo Fisher Scientific). Just before transfection of cell cultures in a 12 well, growth medium was removed, and 0.4 ml of Opti-MEM and the RNA-containing transfection mix were added. After about 5–6 h, the medium was replaced with 0.6 ml of fresh regular culture medium. Cultures were grown for 40–46 h, until confluency was reached and no mitotic cells were noticed under the microscope. At this stage, cells were either collected (t = 0) or treated as indicated. For starvation, cultures were briefly washed with glucose and serum-free medium and replenished with GS starvation medium. GS starvation medium consists of no-glucose Eagle's Minimum Essential Medium with Earle's salts and nonessential amino acids (MCP-202-10L; SERENA), supplemented with 2.6 mM L-glutamine, antibiotics, and 20 mg/liter D-glucose, representing 2% of the glucose concentration in regular culture medium. Starvation durations are indicated in the relevant figures. Chloroquine (C6628; Sigma-Aldrich), bafilomycin A1 (BML-CM110-

0100; Enzo), SAR405 (533063; Sigma–Aldrich), and Torin-1 (inh-tor1; InvivoGen) were added to the fresh medium at the onset of the treatment. For RNA extraction, cells were washed twice with cold PBS (without calcium and magnesium) and harvested, and upon RNA analysis at the 12-well format, cells were exposed to QIAzol reagent (see below) and further processed. For protein analysis, cell pellets were resuspended in 1.5x protein sample buffer (PSB) (50 ml of 4.5 × PSB containing 15 ml glycerol, 7.5 ml $\beta$-mercaptoethanol, 22.5 ml sodium dodecyl sulfate [20%], 6.3 ml Tris buffer, pH = 6.8 [1M], and bromophenol blue).

### Subcellular fractionation

Subcellular fractionation to separate a lysosome-enriched fraction from the cytosol was performed by employing the Sigma–Aldrich LYSISO1 Lysosome Isolation kit (Kacal & Vakifahmetoglu-Norberg, 2022), and according to the manufacturer's instructions. Briefly, cells were harvested and resuspended in the extraction buffer supplemented with the protease cocktail and the RNase inhibitor RNasin. After lysis, serial centrifugation steps were performed saving the cytosol, and after the lysosome-containing fraction was washed and resuspended in extraction buffer. The collected fractions were then subjected to RNA purification.

### Reverse transcriptase quantitative PCR (RT–qPCR)

A miRNeasy kit (217004; QIAGEN) with DNase (79254; QIAGEN) was used to purify cellular RNA, including miRs, according to the manufacturer's protocol. For mRNA expression analysis, reverse transcription was performed with M-MLV Reverse Transcriptase (M1701; Promega), random hexamers, and 0.3–0.5 μg total RNA. Total cDNA (8–10 ng RNA equivalent) was taken for real-time PCRs in the presence of SYBR Green, performed in a StepOnePlus thermocycler with the following cycling protocol: preheating at 95°C for 1":10", followed by 40 two-step cycles of 95°C for 5" and 60°C for 40". Primer sequences are listed in Table S5. RPL8 was used as a normalizing gene, based on its steady expression under our experimental conditions (Michael et al, 2023). Relative quantification was derived using standard curves present in every plate and for each set of primers. qPCR results were considered valid only when amplification plots were within the dynamic range and met all other criteria set by the MIQE guidelines (Bustin et al, 2010). For miRNA expression analysis, 200 ng total RNA was taken for polyA tail addition followed by cDNA synthesis using Quanta Biosciences 95107-025 qScript miRNA cDNA Synthesis Kit according to the manufacturer's instructions (Mestdagh et al, 2014). For first-strand cDNA synthesis and for real-time PCR analysis of relative miRNA expression, PerfeCTa universal primer and a PerfeCTa assay primer were used together with PerfeCTa SYBR Green SuperMix (95054; Quanta Biosciences). Upon pre-amplification capable of generating miR-4488–derived amplicon, primers were used together with primers that generate the SNORD44 endogenous control amplicon. In miRNA RT–qPCR assays, SNORD44 was used as a normalizing gene (Michael et al, 2023). qPCR after pre-amplification was done in a StepOnePlus thermocycler with the following cycling protocol: preheating at 95°C for 2', followed by 40 two-step cycles of 95°C for 4" and 69.5°C for 40". For miRNA qPCR

analysis of expression, we employed the ΔΔCt method; data were retrieved only after measurements were taken to ensure a linear dynamic range.

### Quantification of miR-4488 copies per cell

To quantify miR-4488 copy number per cell in HFFs under basal conditions, total RNA was extracted and subjected to poly-adenylation followed by cDNA synthesis. In parallel, the same procedure was applied to exogenous miR-4488 standards, taking into account that one nanogram of miR-4488 is calculated to contain $8.3 \times 10^{10}$ copies. A practical standard curve range was generated by serial dilution of the cDNA derived from the exogenous miRNA, using yeast tRNA as a carrier. The miR-4488 copy-number values were extracted from the standard curve according to the MIQE guidelines (Bustin et al, 2010). To deduce the number of miR-4488 copies per cell, we used the total RNA amounts taken for cDNA preparation and the cell numbers from which they were derived.

### Aligning miR-4488 to the human ribosome structure

The location of the human extension segment 7L (ES7L) in the large subunit 28S ribosomal RNA was depicted in the 1.67 Å cryo-EM structure of the ribosome. The atomic coordinates and EM map were derived from PDB-8GLP and EMD-40205. The secondary structure prediction of pre-miR-4488 derived from the 28S ribosomal RNA was derived using RNA fold (http://rna.tbi.univie.ac.at/).

### Analysis of the 3′-ends of endogenous miR-4488 using a cDNA library

The first- and second-strand cDNA primer and the primers used for generating the library amplicons are listed in Table S6. Recovered miRNAs were polyadenylated using and cDNA was generated using the qScript Flex cDNA synthesis kit (Quanta) following the manufacturer's gene-specific protocol. A single-step second-strand PCR was performed, and the resulting product was purified using a 2:1 ratio of Sera-Mag beads (GE). The resulting DNA was amplified with primers containing the required sequences for Illumina sequencing. Libraries were sequenced using the Illumina NovaSeq X instrument. Reads were trimmed using Cutadapt (Martin, 2011) with 5′ adaptor = ^HNNHNNAGGGGGCGGGCTCCGG and 3′ adaptor = AAAAAAAAAAAAAAAA, and unique sequences were extracted. UMIs with the same sequence were collapsed and counted. An additional round of cleaning was performed by removing bases at the 3′ if they appear after at least 3 A's. The A's were also removed. The UMIs were collapsed and counted again, and then, UMIs were classified. About 20% of UMIs could not be assigned to any genomic sequence, probably owing to modifications after transcription or to technical issues. The rest of the sequences were of chromosome 11, ribosomal RNA, or either origin.

### Expression array analysis and analysis of high-throughput data

For the mRNA expression array analysis on Clariom S human arrays (902917; Thermo Fisher Scientific), 100 ng total RNA derived from

miR-CTRL, or miR-4488–transfected HFFs, was used. Standardized arrays were processed with Gene Chip WT PLUS reagent kit (902281; REF). Microarray data statistical analysis was performed using Partek Genomics Suite software (Partek Inc.). CEL files (containing raw expression measurements) were imported to Partek GS. Processing and normalization of data was done using the robust multichip average algorithm (RMA) (Irizarry et al, 2003). One-way analysis of variance was applied to assign differentially expressed genes. Contrasts were calculated for each of the three pairwise comparisons. False discovery rate was used to correct for multiple comparisons using the procedure of Benjamini and Hochberg (Benjamini & Hochberg, 1995). Probe sets whose normalized expression intensity was below 5.5 in all the arrays were considered as not detected and were filtered out. $Log_2$-transformed normalized intensity values were used for hierarchical clustering. Samples were clustered using Pearson's dissimilarity distance measure and Ward's linkage. For Heatmap of Partitioning Clustering in the first expression array experiment, genes exhibiting an absolute fold change of at least 2 and false discovery rate (FDR) of 0.05 or less in any pairwise comparison were chosen. K-means partitioning clustering was performed on $log_2$ intensities with Pearson's dissimilarity as a distance measure. For visualization, the $log_2$ intensities were standardized to have for each gene zero mean and unit SD. Standardized $log_2$ intensities are indicated by a colored bar.

Functional analysis of the differentially expressed genes was performed using Ingenuity Pathway Analysis: (https://www.qiagenbioinformatics.com/products/ingenuity-pathway-analysis/).

## Western blot analysis

Protein extracts in PSB (see above) were boiled for 3 min and separated by SDS–PAGE (10% polyacrylamide gel), and proteins were transferred onto a nitrocellulose membrane (Protran BA-83; Whatman). Membranes were reacted with anti-GAPDH (MAB-374; Millipore) anti-ATF4 (D4B8, 11815; Cell Signaling), and IRE1 (14C10, # 3294; Cell Signaling) primary antibodies diluted in 2% BSA in PBST (PBS without calcium and magnesium plus 0.05% Tween-20). After incubation with the appropriate secondary antibodies (Peroxidase AffiniPure Goat Anti-Mouse, 115-035-146, or Peroxidase AffiniPure Goat Anti-Rabbit, 115-035-144, both from Jackson Laboratories), membranes were immersed in enhanced chemiluminescent (ECL) substrate solution (#34080; SuperSignal West Pico, Thermo Fisher Scientific). Digital images were captured using myECL imager (Thermo Fisher Scientific).

## Biotinylated (bt)-miR-mRNA pulldown (PD) experiments

The PD assay was performed to enrich for high-specificity miRNA targets (Tan & Lieberman, 2016; Michael et al, 2023). In short, an HFF cell culture in a 10-cm dish was transfected with 30 nM biotinylated miRNA mimic (bt-miRNA), biotinylated at the 3′-end of the active strand. After 48 h, cells were either harvested (t = 0) and processed or starved for 48 h before harvesting. Streptavidin Dynabeads M-280 (11205D; Invitrogen/Thermo Fisher Scientific) captured mRNA bound to the transfected bt-miR. mRNAs bound to the bt-miRNA-

4488 were compared with those bound nonspecifically in a control pulldown assay with control bt-miRNA mimic (bt-miR-CTRL). Captured RNAs were quantified by RT–qPCR. To increase the sensitivity and reproducibility of the assay, we introduced an RT–pre-amplification step before final qPCR. Specifically, SuperScript III One-Step RT–PCR system (12574-018; Invitrogen/Thermo Fisher Scientific) was employed to generate cDNA and to carry a linear co-amplification (11–14 cycles) of three amplicons: two derived from target genes (final concentration of each primer was 80 nM) and one derived from the RPL8 reference gene (final concentration of each primer was 60 nM). Pre-amplification samples were treated with exonuclease I (NEB) to remove leftover primers. Standard curves for each pair of primers were made, and qPCR results were considered valid only when amplification plots were within the dynamic range and met all other criteria set by the MIQE guidelines (Bustin et al, 2010). qPCR data analysis in PD experiments also involved a couple of normalization steps, to account for differences in target abundance because of bt-miRNA biological effect and for irrelevant technical variation throughout the procedure. Thus, the enrichment ratio for bt-miRNA–bound mRNA was calculated as: (target gene PD/reference gene PD)/(target gene input)/(reference gene input).

## Luciferase assay

To generate plasmids in which either wt or mutated NFKB2 3′UTR sequences are fused downstream of the luciferase coding sequence, we performed sequential cloning as follows. Addgene plasmid #14715 was used as the initial backbone. The GFP coding sequence was removed, and the promoter was replaced with the PGK promoter. Codon-optimized firefly luciferase (LUC2) was amplified from pGL4.13 and inserted into the modified backbone. Finally, the WPRE was deleted, and either the WT NFKB2 3′UTR or a mutant 3′UTR was cloned downstream of the luciferase coding sequence. The first 40 nucleotides of the mutant NFKB2 sequence are CCTGCTGCCTGCTTACAGCTTACTTCCCGGACTTACTGTA. After miRNA transfection (50 nM) in a 12-well format, 40 ng/well of each recombinant reporter plasmid was cotransfected with 40 ng/well of pCIneo-RL (#115366; Addgene), which encodes humanized Renilla luciferase, with 0.9 $\mu$g carrier plasmid using TransIT-2020 Transfection Reagent (MIR 5400; Mirus Bio) according to the manufacturer's instructions. For luminescence quantification, the Promega Dual-Glo Luciferase Assay System reagents were used. Readings were performed using a microplate luminometer (Veritas).

## Pulldown of Argonaute protein complexes and associated microRNAs using the AGO-APP protocol

The "Ago Affinity Purification by Peptides" (Ago-APP) protocol (Hauptmann et al, 2015; Hauptmann & Meister, 2017) was employed to probe for miR-4488 association with Argonaute proteins.

FLAG-GST-T6B WT and mutant peptides were expressed and purified as previously described (Hauptmann et al, 2015). In short, proteins were expressed in BL21-Gold(DE3)pLysS–competent cells. Bacteria were grown at 18°C to OD 0.6, induced with 1 mM isopropyl b-D-1-thiogalactopyranoside (IPTG) (R0392; Thermo Fisher

Scientific), and harvested at OD 1.3–1.4. Bacterial pellets were resuspended in GST-A lysis buffer (PBS containing 10% lysosome 10 mg/ml (10 837 059 001; Roche) and 1 mM DL-dithiothreitol (DTT) (43815; Sigma-Merck) and HALT protease inhibitor cocktail (78438; Thermo Fisher Scientific)) followed by three rounds of sonication for 3 min at 100% amplitude (VCX130; Sonics). Lysates were cleared for 20 min by centrifugation at 20,000$g$. The cleared lysates were loaded onto columns containing 2 ml of bead volume glutathione Sepharose beads (L00206; A2S) and washed two times with GST-A buffer. The GST-tagged WT T6B and mut T6B were eluted in freshly made 10 ml of GST-B buffer (20 mM Tris, pH 8.0, and 10 mM glutathione [G4251; Sigma-Aldrich] in PBS). The fusion peptides were concentrated using Amicon Ultra-15 Centrifugal Filter Unit (Millipore).

For microRNA pulldown by Ago-APP, we merged two published protocols (Gagliardi & Matarazzo, 2016; Su et al, 2022), with some additional modifications. Cells in 10-cm dishes were washed twice with 3 ml PBS and harvested with 1 ml trypsin, and the cell pellets (obtained by centrifuging at 300$g$ for 5 min at 4°C) were washed twice with PBS. Each pellet was then resuspended in ~150 $\mu$l GS lysis buffer (150 mM KCl, 5 mM MgCl2, 50 mM Tris 7.4, 0.5% NP-40, and 0.5 mM DTT [all reagents were at RNA grade]), freshly supplemented with protease inhibitors cocktail (539134-1ML; Calbiochem), phosphatase inhibitor cocktail (524625; Calbiochem), and RNasin as an RNase inhibitor (N2611; Promega). Pellets were resuspended by pipetting up and down, vortexed, and incubated on ice for 5 min. To further promote cell lysis, pellets were frozen at –80°C *before pipetting*. To prepare the FLAG antibody resin, 75 $\mu$l FLAG gel (A2220; Sigma-Aldrich) per immunoprecipitation was washed twice in PBS-Gly-DTT-PI buffer (PBS without calcium and magnesium, 5% glycerol, 0.5 mM DTT [43815; Sigma-Aldrich] supplemented with protease inhibitors) at 4,000$g$ for 1 min. T6B proteins (typically for three batches) were applied in 400 $\mu$l PBS-Gly-DTT-protease inhibitors. FLAG resins were incubated and rotated at 4°C for 3–4 h in the presence of the T6B peptides. The resin was then washed twice in PBS-Gly-DTT-protease inhibitors and then once with IP buffer (50 mM Tris 7.4, 150 mM NaCl, 1 mM MgCl2, 0.5% NP-40, glycerol 5%, supplemented with protease inhibitors, phosphatase inhibitors, and RNasin). Frozen lysates were then thawed and centrifuged at 20,000$g$ for 10 min at 4°C. Supernatants were collected, and 10% of each supernatant was taken as "input" and immediately supplemented with RNA lysis buffer, QIAzol. About 120 $\mu$l of the supernatant was applied to the antibody-gel resin and rotated for 4 h at 4°C. Pulled-down material was washed five times with NET buffer (50 mM Tris 7.4, 150 mM NaCl, 5 mM EDTA, 0.5% NP-40, 10% glycerol) and once with PBS. QIAzol was applied to the resin, and RNA was prepared as described in the Materials and Methods section. RNA from the pulled-down material (the immunoprecipitated material) and the input samples were probed for the relative amounts of SNORD44 (as a reference RNA) and miR-4488 either with or without prior pre-amplification followed by qPCR. To calculate relative miR-4488 quantities, miR-4488/SNORD44 ratio in the input samples was used to divide miR-4488/SNORD44 ratios derived from the pulled-down samples in order to normalize and correct for all possible technical anomalies.

### Statistical analysis

For statistical analysis, and unless otherwise stated, data were derived from three biological replicates. Statistical analysis of qPCR results was performed with Partek Genomics Suite software (Partek Inc.) or using the "aov" function in R. Log$_2$-transformed relative expression values and ΔCt values were imported for mRNA and miRNA statistical analysis, respectively. Contrasts were calculated to compare pairwise between the given parameters. FDR was used to correct for multiple comparisons. Unless otherwise stated, relevant FDR values are presented in the Supporting Statistical Analysis data. NSS or ns stands for not statistically significant. Heatmaps were drawn using Excel's conditional formatting. The graphs were produced using the ggplot2 package in R. The values shown in the plots were batch-corrected in R using a linear model. Error bars represent the SD. The q-values displayed in the graphs represent FDR values.

## Data Availability

Relevant data that support the findings of this study are available in Supplementary Material. The microarray data that support the findings of this study are available in NCBI's Gene Expression Omnibus and are accessible through GEO accession GSE244149.

## Supplementary Information

## Acknowledgements

We would like to thank Ido Zur, Evyatar Shaked, and Benjamin Stampfer for technical assistance with some of the molecular biology experiments and Ishai Sher, Gal Ambar, Genia Brodsky, and Sahlav Bar for assistance with graphical design. We are indebted to Gunter Meister for providing the plasmids encoding GST-T6B proteins. We appreciate stimulating discussions with Aishe Sarshad and the help provided by Shira Albeck in the purification of the T6B proteins. This work was supported in part by the Dr. Miriam and Sheldon G. Adelson Medical Research Foundation and the Moross Integrated Cancer Center.

### Author Contributions

D Michael: conceptualization, data curation, formal analysis, supervision, validation, investigation, visualization, methodology, project administration, and writing—original draft, review, and editing.
E Feldmesser: data curation, software, formal analysis, validation, visualization, methodology, and writing—review and editing.
K S Rajan: data curation, formal analysis, investigation, visualization, methodology, and writing—review and editing.
Y Lubelsky: data curation, software, investigation, and methodology.
A Bashan: data curation, software, formal analysis, visualization, methodology, and writing—review and editing.

A Damalas: conceptualization and methodology.

S Ben Zvi: investigation.

A Friedberg: investigation.

N Eitan: investigation.

S Erez: investigation.

A Yonath: conceptualization and methodology.

I Ulitsky: conceptualization, software, supervision, methodology, and writing—review and editing.

M Oren: conceptualization, resources, supervision, funding acquisition, validation, methodology, and writing—review and editing.

## Conflict of Interest Statement

The authors declare that they have no conflict of interest.

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
