## [Reviewer comments · Life Science Alliance]

A microRNA generated via lysosomal processing of ribosomal RNA suppresses proinflammatory responses

Dan Michael, Ester Feldmesser, K. Shanmugha Rajan, Yoav Lubelsky, Anat Bashan, Alexandros Damalas, Shiri Ben Zvi, Aya Friedberg, Netta Eytan, Shachar Erez, Ada Yonath, Igor Ulitsky, and Moshe Oren

DOI: <https://doi.org/10.26508/lsa.202503536>

Corresponding author(s): Moshe Oren, Weizmann Institute of Science and Dan Michael, The Weizmann Institute of Science

Review Timeline:

Submission Date:	2025-10-15
Editorial Decision:	2025-11-07
Revision Received:	2026-02-21
Editorial Decision:	2026-04-01
Revision Received:	2026-04-21
Accepted:	2026-04-21

Scientific Editor: Tim Fessenden

Transaction Report:

November 7, 2025

Re: Life Science Alliance manuscript #LSA-2025-03536-T

Moshe Oren
The Weizmann Inst. of Science
Department of Molecular Cell Biology
The Weizmann Institute of Science
POB 26
Rehovot, Rehovot 76100
Israel

Dear Dr. Oren,

Thank you for submitting your manuscript entitled "A microRNA generated via lysosomal processing of ribosomal RNA suppresses proinflammatory responses" to Life Science Alliance. The manuscript was assessed by expert reviewers, whose comments are appended to this letter.

As you will see, reviewers expressed interest in the identification of this microRNA and the novel processing mechanism proposed. However, all reviewers requested validation, controls, and additional observations to support your model. Namely, Reviewers 1 and 3 noted concerns on the use of 293T and HCT116 cells, in the absence of stressors, to examine Dicer and Drosha and requested using fibroblasts under stress for these assays. Reviewers 1 and 2 both remarked that miR-4488 targets shown in Fig 8 should be validated with a Luciferase assay. Reviewer 2 requested confirmation and greater details on miR-4488 processing by autophagic/lysosomal flux, and we agree that these observations should be strengthened in the manner of your choice. Reviewers also sought important controls and validation of KO efficiency. Additional experimental data beyond those mentioned here are not required in a revision, however a rebuttal must respond to all reviewer points in some manner.

I would be happy to discuss the revision in more detail via email or phone/videoconferencing. Please let me know which option you prefer, if any.

While you are revising your manuscript, please also attend to the below editorial points to help expedite the publication of your manuscript. Please direct any editorial questions to the journal office. When submitting the revision, please include a letter addressing the reviewers' comments point by point.

Thank you for this interesting contribution to Life Science Alliance. I hope that the comments below will prove constructive as your work progresses, and I are looking forward to receiving your revised manuscript.

Sincerely,

- A letter addressing the reviewers' comments point by point.
- An editable version of the final text (.DOC or .DOCX) is needed for copyediting (no PDFs).
- High-resolution figure, supplementary figure and video files uploaded as individual files: See our detailed guidelines for

preparing your production-ready images, <https://www.life-science-alliance.org/authors>

B. MANUSCRIPT ORGANIZATION AND FORMATTING:

Reviewer #1 (Comments to the Authors (Required)):

Reviewer comments to Michael et al: "A microRNA generated via lysosomal processing of ribosomal RNA suppresses proinflammatory responses".

Summary

The authors identify miR-4488 as induced in fibroblasts during glucose and serum starvation. Intriguingly, the production of miR-4488 is independent of Drosha and the pre-miR seems to originate from the ribosomal ES27L. The authors link miR-4488 production to autophagy and the lysosome. Overexpression of miR-4488 down-regulates genes in the proinflammatory response and is speculated to dampen this response.

Specific remarks

Figures 1 and 2:

Fig 1B: The upregulation of miR-4488 is largely driven by a single replicate. In general, the variations in the qRT-PCR data presented in figures 1 and 2 are surprisingly high, maybe indicating that miR-4488 levels are at the border of detection? Can the authors estimate the number of miR-4488 molecules per cell, perhaps using a titration curve? If the absolute levels are very low and the induction only two-fold, can we then expect to see strong regulatory capacity? The dependency of Dicer - but not Drosha - is done in other cell types without starvation induction - given that several miR-4488 genomic locations exist, these experiments should be repeated in the fibroblasts.

Figure 3: The claim that (some parts of) miR-4488 may come from ES7L is interesting and well substantiated experimentally.

Figure 4: Although the authors link the levels of miR-4488 to autophagy using inhibitors, it is unclear how this would work? The authors suggest "initial processing in the lysosome followed by Dicer processing in the cytoplasm". It is quite obscure to this reviewer which mechanisms would be involved, and the authors should substantiate this further. Alternatively, blocking autophagy induces cellular stress and the generation of miR-4488 could be a secondary effect.

Figure 6: How is miR-4488 generation regulated downstream starvation? It appears from figure 6 that overexpressed miR-4488 is only engaged in target regulation following starvation and that the miR-4488 has no effect in the absence of starvation? The way this heatmap is normalized is confusing with CTRL to T0 and miR-4488 to CTRL. Also, the authors should show the efficiency of miR-4488 inhibition.

Figure 7: The authors should include the miR-4488 inhibitor in these experiments.

Figure 8: A standard luciferase reporter assay with target regions +/- mutations in the suggested target regions should be performed. Comparing 8B and 8C, there does not seem to be much difference for RELB?

Reviewer #2 (Comments to the Authors (Required)):

Authors present a non-canonical microRNA, miR-4488, arising from the ES7L expansion segment of human 28S rRNA suppress ER-stress-induced inflammatory programs. They show Drosha-independent yet Dicer-dependent biogenesis and Argonaute association of miR-4488 using loss-of-function genetics and Ago pulldown experiments, respectively. Deep, targeted sequencing of endogenous molecules demonstrates that the majority of miR-4488 reads map to rRNA loci rather than the annotated chr11 genomic site, with the mature sequence alignment to 28S positions and to a predicted pre-miRNA-like hairpin in ES7L. The production of miR-4488 requires the autophagosome-to-lysosome axis: inhibitors of VPS34 (SAR405), v-ATPase (Bafilomycin A1) and lysosomal degradation (Chloroquine) suppressed its production, whereas Torin1 (mTORC1/2 inhibitor) induced it. Functionally, ectopic miR-4488 attenuates pro-inflammatory cytokines/chemokines and UPR drivers during glucose/serum (GS) starvation, while a hairpin inhibitor has the opposite effect early in stress. At the protein level, ATF4 and IRE1 α are reduced in cells expressing miR-4488 under starvation. They showed that NFKB2 and RELB mRNAs associate with miR-4488 only under stress by the pulldown experiment. However, the mechanistic steps that they proposed remain inferred rather than directly visualized. There is no isolation/imaging of a lysosomal pre-miR-4488 intermediate-and direct, base-pair-level target validation (e.g., via reporter mutagenesis) is not shown for NFKB2/RELB seed sites.

1. Direct evidence of lysosomal processing is needed. Figure 4 argues for pathway involvement, but no lysosomal precursor/pre-miR-4488 species is shown. Please add a figure that demonstrates an alignment of all sequenced reads including processing intermediates with lysosomal RNA-seq under Torin1 {plus minus} Dicer.

2. Please provide a detailed miRNA-target pairing status at nucleotide resolution. Figure 8 shows stress-dependent pulldown of NFKB2/RELB and transcriptional effects, but luciferase reporter assays(WT/mutant of CDS and 3'UTR seed sites) and, ideally, AGO-PAR-CLIP/CLASH to pinpoint crosslinking sites should be done. This would provide a solid direct evidence of targeting in the non-canonical NF- κ B branch.

3. Regarding genomic origin annotation vs. rRNA origin of miR-4488. While miRBase annotates miR-4488 at chr11, this work shows 219 matches across 28S rRNA loci and targeted 3'-extension sequencing where ~99% of {greater than or equal to}19-nt UMIs map to rRNA, not the genomic site-challenging current annotation. It would be great if they present all sequenced reads from 28S rRNAs and chr11 aligned to the hairpin structure to understand the processing intermediates in fig. 3c.

4. Authors did not demonstrate a mechanism how the hairpin structure with a long lower stem like a pri-miRNA is processed into pre-miRNA form (as a dicer substrate), although they claim that the miR-4488 is Drosha-independent product.

- They are mistakenly using GS (glucose and serum) and SG starvation in Methods; make a consistent to GS starvation throughout.
- In page 40, appendix table2: there is typo: starnd strand

Reviewer #3 (Comments to the Authors (Required)):

This manuscript by Michael et al showed how miR-4488 exerts its function during ER stress. The authors identified NF- κ B and RelB as the direct targets of miR-4488, a microRNA with its generation regulated by autophagy-lysosome signaling, and further showed this regulatory axis manipulates ER stress progression at the early stage. These findings are potentially interesting. The following suggestions may help for further improvement.

1. I wonder how miR-4488 was selected? If any other miRNAs cooperate with miR-4488 to regulate ER stress.
2. In Fig 2A, Drosha KO efficiency should be experimentally validated, such as WB.
3. In Fig 2A, I wonder why Drosha KO induces miR-4488 upregulation and if Drosha KO correlates with ER stress.
4. In Fig 2, why the authors investigated how miR-4488 is generated at normal conditions, even though miR-4488 is endogenously expressed at very low level in cells without ER stress as indicated in Fig 6B.
5. In Fig 2, why the authors did not use HFF cells instead of 293T and HCT116 cells.
6. Discussion suggestion: if this regulatory axis identified in this study is specific in mammalian cells? why cells allow such specifically restraining of miR-4488 in a short-time window on NF- κ B signaling-associated inflammation during ER stress.

We thank all the reviewers for their valuable comments and suggestions. We hope that by addressing these comments, as indicated below, the revised manuscript has now been improved.

Reviewer # 1

1. *“Figures 1 and 2:*

Fig 1B: The upregulation of miR-4488 is largely driven by a single replicate. In general, the variations in the qRT-PCR data presented in figures 1 and 2 are surprisingly high, maybe indicating that miR-4488 levels are at the border of detection? Can the authors estimate the number of miR-4488 molecules per cell, perhaps using a titration curve? If the absolute levels are very low and the induction only two-fold, can we then expect to see strong regulatory capacity? The dependency of Dicer - but not Drosha - is done in other cell types without starvation induction - given that several miR-4488 genomic locations exist, these experiments should be repeated in the fibroblasts”.

We have followed the reviewer's valuable recommendation and performed a quantitative qPCR rather than a relative qPCR to estimate the copy number of miR-4488 per cell (see page 5 line 14, and Supplementary materials and methods). Accordingly, we estimated the numbers to be in the range of 2000 copies per cell, not a negligible number even as compared to other microRNAs. Regarding the increase of miR-4488 following starvation, given that there was no difference in the experimental conditions that were used to obtain the results in Fig. 1B and Fig. 4 A-C, we used the data from these 12 replicates for a follow-up analysis and graphical representation, now shown in Fig. S4. We relate to these results in the text in page 8 line 16. As for the dependency of miR-4488 generation on Drosha and Dicer in HFF, we were unable to make progress in this front, mainly for the following reasons. First, we could not perform CRISPR-based experiments in these primary cells as they can be maintained for only 6 mild passages without losing their regular proliferative phenotype. Second, we also expect lethality due to complete KO in these primary cells as even in cancer cells KO of these genes was rarely achieved. Third, knockdown in HFF, our last resort, was not informative: we tried it, but were unable to obtain a sufficient reduction in the expression levels of these genes. In particular, the residual level of Dicer remaining after the knockdown was estimated to be at least 40-50%, and apparently still sufficient to process miR-4488. Therefore, at present we have to rely on the results in HCT116 cells, as in this system all Dicer molecules are compromised to the same level, most likely in a severe manner, unveiling its role in miRNAs maturation. However, in light of the reviewer's important and legitimate concern, we have revised the manuscript and included in the Discussion a comment clarifying that we cannot rule out other modes of miR-4488 generation in primary cells that are Dicer-independent or can supplement Dicer-dependent production of miR-4488 (page 12, line 27).

2. *“Figure 4: Although the authors link the levels of miR-4488 to autophagy using inhibitors, it is unclear how this would work? The authors suggest “initial processing in the lysosome followed by Dicer processing in the cytoplasm”. It is quite obscure to this reviewer which mechanisms would be involved, and the authors should substantiate this further. Alternatively, blocking autophagy induces cellular stress and the generation of miR-4488 could be a secondary effect”.*

To substantiate the role of lysosomes in initiating miR-4488 production, we assessed the presence of the precursor of miR-4488 (pre-miR-4488) using a primer ending just before the first nucleotide of the mature miR-4488 sequence (Fig. S5A). Following cell fractionation, we find this precursor both in the lysosome-containing fraction and in the cytosolic fraction (Fig. S5). The text was revised accordingly (page 8, line 24). In support to this conclusion, we show that blocking the lysosome and independently blocking autophagy results in a substantial decrease in miR-4488 production.

3. *“Figure 6: How is miR-4488 generation regulated downstream starvation? It appears from figure 6 that overexpressed miR-4488 is only engaged in target regulation following starvation and that the miR-4488 has no effect in the absence of starvation? The way this heatmap is normalized is confusing with CTRL to T0 and miR-4488 to CTRL. Also, the authors should show the efficiency of miR-4488 inhibition”.*

To address the reviewer’s concerns, we attempted to further clarify how the heatmap in Figure 6 was normalized. Thus, we split the two-panel image in Fig. 6 into 4 panels (A-D) and expanded the explanation of the normalization in the text (page 9 lines 24). The reviewer is correct in pointing out that from Fig. 6 one learns that the exogenous miR-4488 exerts a substantial part of its activity in a conditional manner, mainly following extended metabolic perturbations. In non-starved cells, the exogenous miR-4488 most likely has no additive value over the substantial endogenous levels of the miRNA, which is still active (see below). We believe that this makes the exogenous miRNA a potential therapeutic tool that could spare cells that are not significantly stressed. As for the regulatory pathway of miR-4488 downstream starvation, under basal non-starved conditions and also at t=4 and t=24, the data indicates that the endogenous miRNA is active (presently panel D in Fig. 6). Unfortunately, estimating miR-4488 inhibition by HP-inhibitors is not feasible as the inhibitor does not reduce the miRNA, but rather blocks its interaction with downstream targets.

4. *“Figure 7: The authors should include the miR-4488 inhibitor in these experiments.”*

The revised Fig. 6D shows the analysis of the levels of IRE1a (encoded by ERN1) and ATF4 mRNAs in cells transfected with the HP-inhibitor.

5. *“Figure 8: A standard luciferase reporter assay with target regions +/- mutations in the suggested target regions should be performed. Comparing 8B and 8C, there does not seem to be much difference for RELB?”*

As requested, we attempted to employ the luciferase assay to analyze the effect of miR-4488 on the predicted sites found in the 3'UTR of NFKB2 mRNA, as the other predicted sites in RELB and NFKB2 mRNA are positioned within the coding sequence. The rationale was that analyzing the effect of miRNAs on sites within the coding sequence employing luciferase chimeric constructs is often non-informative. First, it has to be tested in a coding sequence, under ribosome competing conditions. Furthermore, one cannot freely mutate the sites without risking changes in the functionality of the protein; even in the case of synonymous mutations, mRNA stability might be altered, confounding the ability to draw quantitative conclusions. The results shown in Fig. S6 (addressed in page 24) show that although we could not completely overcome non-specific effects in our assay, possibly due to the combination of weak non-specific binding sites in the non-3'UTR parts of the plasmid and the transfection-induced stress that could limit the physiological effects of miR-4488, a modest but clearly noticeable effect on Firefly luciferase expression was observed and attributed to the specific binding of miR-4488 to the 3'UTR of NFKB2 mRNA. As for RELB, the experiment indeed may hint that RELB mRNA may associate with miR-4488 even at basal conditions, which can still be stressful, but we could not obtain a statistically conclusive result.

Reviewer # 2

“1. Direct evidence of lysosomal processing is needed. Figure 4 argues for pathway involvement, but no lysosomal precursor/pre-miR-4488 species is shown. Please add a figure that demonstrates an alignment of all sequenced reads including processing intermediates with lysosomal RNA-seq under Torin1 {plus minus} Dicer”.

We thank the reviewer for pointing out the need to study the precursor of miR-4488 (pre-miR-4488). To substantiate the role of lysosomes in initiating miR-4488 production, we assessed the presence of the precursor of miR-4488 (pre-miR-4488) using a primer ending just before the first nucleotide of the mature miR-4488 sequence (Fig. S5A). Following cell fractionation, we find this precursor both in the lysosome-containing fraction and in the cytosolic fraction (Fig. S5). The text was revised accordingly (page 8, line 24). In support to this conclusion, we show that blocking the lysosome and independently blocking autophagy results in a substantial decrease in miR-4488 production. Specifically, Torin1 enhances the levels of pre-miR-4488.

Unfortunately, as explained in our reply to Reviewer #1, we were unable to demonstrate the role of Dicer in this pathway in HFF, owing to technical constraints. We have therefore modified the relevant text in the Discussion, as explained in the reply to Reviewer #1.

“2. Please provide a detailed miRNA-target pairing status at nucleotide resolution. Figure 8 shows stress-dependent pulldown of NFKB2/RELB and transcriptional effects, but luciferase reporter assays(WT/mutant of CDS and 3’UTR seed sites) and, ideally, AGO-PAR-CLIP/CLASH to pinpoint crosslinking sites should be done. This would provide a solid direct evidence of targeting in the non-canonical NF-κB branch.”.

Despite our previous failures in such reporter assays in HFF, we gave this approach another chance, calibrating our protocol and among some other parameters also using a new transfection reagent. Thus, we performed a luciferase assay to analyze the effect of miR-4488 on the predicted sites found in the 3’UTR of NFKB2 mRNA(Fig. S6) as the other predicted sites in RELB and NFKB2 mRNA are positioned within the coding sequence, potentially confounding the interpretation of the results (as explained in our reply to the last comment of Reviewer #1). The results shown in Fig. S6 (addressed in page 24) show that although we could not completely overcome non-specific effects in our assay, possibly due to the combination of weak non-specific binding sites in the non-3’UTR parts of the plasmid and the transfection-induced stress that could limit the physiological effects of miR-4488, a modest but clearly noticeable effect on Firefly luciferase expression was observed and attributed to the specific binding of miR-4488 to the 3’UTR of NFKB2 mRNA

“3. Regarding genomic origin annotation vs. rRNA origin of miR-4488. While miRBase annotates miR-4488 at chr11, this work shows 219 matches across 28S rRNA loci and targeted 3’-extension sequencing where ~99% of {greater than or equal to}19-nt UMIs map to rRNA, not the genomic site-challenging current annotation. It would be great if they present all sequenced reads from 28S rRNAs and chr11 aligned to the hairpin structure to understand the processing intermediates in fig. 3c.”.

We share the reviewer’s view that the data presented in Fig. 3D should be extended to highlight the different miR-4488 variants derived from the ribosomal 28S RNA. We therefore added Fig. S2, marking the different 3’-ends of the miR-4488 variants. Positions 19 to 24 are unique to the ribosomal RNA and are not found in chromosome 11 as an extension of the 18-nucleotide core sequence.

“4. Authors did not demonstrate a mechanism how the hairpin structure with a long lower stem like a pri-miRNA is processed into pre-miRNA form (as a dicer substrate), although they claim that the miR-4488 is Drosha-independent product.”.

We thank the reviewer for pointing out this highly relevant issue. We can only say that at this point we don’t have a definitive answer, but we plan to address this issue comprehensively in the future, beyond the scope of the present study, following some preliminary leads.

<Minor comments>

- They are mistakenly using GS (glucose and serum) and SG starvation in Methods; make a consistent to GS starvation throughout.
- In page 40, appendix table2: there is typo: starnd à strand

Thank you, the requested corrections were made.

Reviewer # 3

1. *I wonder how miR-4488 was selected? If any other miRNAs cooperate with miR-4488 to regulate ER stress.*

In response to the reviewer's question, the text was revised as follows: "Based on our previous microarray analysis {Michael, 2023} in the present study we focused on another miRNA whose levels, similar to miR-4734, changed upon 48 hours of starvation, miR-4488." miR-4734 is also known to regulate ER stress and the proinflammatory response, with some differences that are discussed in the text.

2. *In Fig 2A, Drosha KO efficiency should be experimentally validated, such as WB.*

The Drosha KO was done by the group of David A. Williams. The relevant paper (Park et al. 2018) includes also a Western analysis.

3. *"In Fig 2A, I wonder why Drosha KO induces miR-4488 upregulation and if Drosha KO correlates with ER stress."*

We concur with the hypothesis raised by the reviewer. However, since we are focusing on physiological settings in primary cells, we did not further investigate 293 cells.

4. *In Fig 2, why the authors investigated how miR-4488 is generated at normal conditions, even though miR-4488 is endogenously expressed at very low level in cells without ER stress as indicated in Fig 6B.*

As the reviewer points out, we indeed find that the endogenous miR-4488 is active in HFF under basal non-starved conditions. We believe that this may reflect some basal cell culture-induced stress. Similarly, the 293 and HCT116 cells may also experience some tissue culture-induced stress, particularly since they are highly proliferative and very active metabolically, thus likely consuming rapidly ingredients of the culture medium.

5. *In Fig 2, why the authors did not use HFF cells instead of 293T and HCT116 cells.*

We agree with the reviewer that ideally this should have been performed in HFF. However, as explained in our reply to comment 1 of Reviewer #1, this turned out to be technically problematic. However, in light of the reviewer's legitimate concern, we included in the Discussion a comment that clarifies that we cannot rule out other modes of miR-4488

generation in primary cells that are Dicer-independent or can supplement Dicer-dependent production of miR-4488 (page 12, line 27).

6. Discussion suggestion: if this regulatory axis identified in this study is specific in mammalian cells? why cells allow such specifically restraining of miR-4488 in a short-time window on NF- κ B signaling-associated inflammation during ER stress.

We believe that miR-4488 acts in a short time window to restrain spurious cytokine and chemokine response and to prevent inflammation in case the stress conditions turn to be transient rather than chronic.

We thank the reviewer for her/his suggestion, and the Discussion section was modified accordingly (page 13, line 19): "Therefore, it is likely that the transient action of miR-4488 allows cells to distinguish between relatively harmless time-limited stress conditions and other scenarios that involve progressive and extended stress, the latter necessitating a proinflammatory response."

April 1, 2026

RE: Life Science Alliance Manuscript #LSA-2025-03536-TR

Prof. Moshe Oren
Weizmann Institute of Science
Department of Molecular Cell Biology
The Weizmann Institute of Science
POB 26
Rehovot, Rehovot 76100
Israel

Dear Dr. Oren,

Thank you for submitting your revised manuscript entitled "A microRNA generated via lysosomal processing of ribosomal RNA suppresses proinflammatory responses". We appreciate your patience during the review process, which was delayed by reviewer availability. As you will see, Reviewer 3 is now satisfied with no further requests. We would be happy to publish your paper in Life Science Alliance pending final revisions necessary to meet our formatting guidelines.

MANUSCRIPT ORGANIZATION AND FORMATTING:

To avoid unnecessary delays in the acceptance and publication of your paper, please read the following information carefully. Full guidelines are available on our Instructions for Authors page, <https://www.life-science-alliance.org/authors>

- Please upload your main and supplementary figures as single files.
- Please add the X and Bluesky handles of your host institute/organization, as well as your own, and/or one of the authors, in our system.
- Please upload a clean manuscript file without the track changes. The one that highlights the changes you can upload as "Related Manuscript File".
- Please be sure that the authorship listing and order are correct and match between the system and the manuscript file.
- Please add your main, supplementary figure, and table legends to the main manuscript text after the references section.
- Please consult our manuscript preparation guidelines <https://www.life-science-alliance.org/manuscript-prep> and make sure your manuscript sections are in the correct order.
- Please rename "Competing interests" to Conflict of Interests.
- Please be sure that all authors are mentioned in the Authors' Contribution section in the manuscript file.
- The contributions selected for Moshe Oren and Ada Yonath do not qualify them for authorship. Please either update the contributions in our system and in the Author Contributions section of the manuscript, or let us know if the authors need to be removed (and added eventually to the acknowledgment section).
- We encourage you to revise the figure legend for Figure 1 such that the figure panels are introduced in alphabetical order.
- Please add a callout for Figure 1A to your main manuscript text.
- LSA articles do not include supplementary methods. Please move the methods from this section to the main Materials and Methods section.
- Please remove the reviewer token from the Data Availability section.
- Please rename all tables as Supplementary Table files.

We welcome submissions of potential cover images for the issue of LSA in which your work would appear. If you have high quality images associated with this work, please feel free to email these, with a caption, to the journal office.

LSA encourages authors to provide a 30-60 second video where the study is briefly explained. We will use these videos on social media to promote the published paper and the presenting author (for examples, see <https://docs.google.com/document/d/1-UWCfbE4pGcDdcgzcmiuJI2XMBJnxKYeqRvLLrLSo8s/edit?usp=sharing>). Corresponding or first-authors are welcome to submit the video. Please submit only one video per manuscript. The video can be emailed to contact@life-science-alliance.org

FINAL FILES:

The following items are required for acceptance.

The license to publish form must be signed before your manuscript can be sent to production. A link to the license to publish form will be available to the corresponding author only. Please take a moment to check your funder requirements.

Thank you for your attention to these final processing requirements. Please revise and format the manuscript and upload materials as soon as you are able.

Thank you for this interesting contribution to the literature. We look forward to publishing your paper in Life Science Alliance.

Sincerely,

Reviewer #3 (Comments to the Authors (Required)):

This revised version is much better than previous one. And all my concerns have been addressed.

April 21, 2026

RE: Life Science Alliance Manuscript #LSA-2025-03536-TRR

Prof. Moshe Oren
Weizmann Institute of Science
Department of Molecular Cell Biology
The Weizmann Institute of Science
POB 26
Rehovot, Rehovot 76100
Israel

Dear Dr. Oren,

Thank you for submitting your Research Article entitled "A microRNA generated via lysosomal processing of ribosomal RNA suppresses proinflammatory responses". It is a pleasure to let you know that your manuscript is now accepted for publication in Life Science Alliance. Congratulations on this interesting work.

Your article will publish open access upon publication under a CC-BY license.

DISTRIBUTION OF MATERIALS:

Again, congratulations on a very nice paper. I hope you found the review process to be constructive and are pleased with how the manuscript was handled editorially. We look forward to future exciting submissions from your lab.

Sincerely,
